# Clinical Outcomes of EUS-Guided Choledochoduodenostomy for Biliary Drainage in Unresectable Pancreatic Cancer: A Case Series

**DOI:** 10.3390/medicina59020351

**Published:** 2023-02-13

**Authors:** Bozhidar Hristov, Deyan Radev, Petar Uchikov, Gancho Kostov, Mladen Doykov, Siyana Valova, Eduard Tilkiyan

**Affiliations:** 1Second Department of Internal Diseases, Section “Gastroenterology”, Medical Faculty, Medical University of Plovdiv, 6000 Plovdiv, Bulgaria; 2Gastroenterology Clinic, University Hospital “Kaspela”, 4001 Plovdiv, Bulgaria; 3Department of Special Surgery, Faculty of Medicine, Medical University of Plovdiv, 6000 Plovdiv, Bulgaria; 4St. George University Hospital, 4000 Plovdiv, Bulgaria; 5Department of Surgery, University Hospital “Kaspela”, 4001 Plovdiv, Bulgaria; 6Department of Urology and General Medicine, Medical Faculty, Medical University of Plovdiv, 6000 Plovdiv, Bulgaria; 7Clinic of Urology, University Hospital “Kaspela”, 4001 Plovdiv, Bulgaria; 8Second Department of Internal Diseases, Section “Nephrology”, Medical Faculty, Medical University of Plovdiv, 6000 Plovdiv, Bulgaria; 9Clinic of Nephrology, University Hospital “Kaspela”, 4001 Plovdiv, Bulgaria

**Keywords:** pancreatic cancer, endoscopic ultrasound, biliary drainage, choledochoduodenostomy

## Abstract

*Introduction.* Pancreatic ductal adenocarcinoma (PDA) is associated with poor prognosis and 98% loss-of-life expectancy. 80% of patients with PDA are unfit for radical surgery. In those cases, emphasis is set on management of cancer-related symptoms, among which obstructive jaundice is most common. Endoscopic ultrasound-guided biliary drainage (EUS-BD) emerges as a valid alternative to the well-accepted methods for treatment of biliary obstruction. *Patient Selection.* Five consecutive patients with unresectable pancreatic malignancy, were subjected to EUS-BD, particularly EUS-guided choledochoduodenostomy (EUS-CDS). *Ethics.* Oral and written informed consent was obtained in all cases prior procedure. *Technique.* EUS-guided puncture of the common bile duct was performed, followed by advancement of a guidewire to the intrahepatic bile ducts. After dilation of the fistulous tract with a cystotome, a fully covered self-expandable metal stent was inserted below the hepatic confluence and extending at least 3 cm in the duodenum. Technical and clinical success was achieved in four patients without adverse events. In one patient procedure failed due to dislocation of the guidewire, with consequent biliary leakage requiring urgent surgery. Recovery was uneventful with no further clinical sequelae and there was no mortality associated with procedure. *Discussion.* Introduced in 2001, EUS-guided biliary drainage has become an accepted option for treatment of obstructive jaundice. According to recent guidelines published by European Society of Gastrointestinal Endoscopy (ESGE) in 2022, EUS-CDS is a preferred modality to percutaneous transhepatic biliary drainage (PTBD) and surgery in patients with failed ERCP, with comparable efficiency and better safety profile, which is supported by our experience with the procedure. *Conclusions.* Our case series suggests that EUS-CDS is an excellent option for palliative management of malignant distal biliary obstruction, emphasizes on the importance of adequate technique and experience for the technical success, and urges the need for future research on establishing the best choice for guidewire and dilation device.

## 1. Introduction

Pancreatic cancer represents a significant burden for the contemporary medicine. Though placed seventh in malignant disease incidence, it is the fourth leading cause of cancer-related death in Europe and third leading in the USA [1,2]. By 2030, PDA is expected to become the second most common cause for death due to malignant disease and by 2035 the incidence is believed to rise by 50% [3]. In spite of the development of medical science, the prognosis in patients with pancreatic cancer has not improved in the last 20 years, the disease being associated with 98% loss-of-life expectancy [4]. Unfortunately, at the time of diagnosis 80% of patients are considered unfit for curative surgery, while in the remaining 20% subjected to radical surgery, local or metastatic recurrence is established during follow-up. In patients with locally advanced and metastatic disease, the emphasis is set on management of cancer related symptoms and improvement of life quality.

Obstructive jaundice is established in about 70% of the patients with pancreatic cancer, being one of the most common clinical manifestations. Currently, the standard approach in those cases is endoscopic stenting by means of endoscopic retrograde cholangiopancreatography (ERCP). Although generally safe (5.18–9.8% complication rate) and with high success rate, in about 3–10% of the patients it is impossible to achieve biliary drainage [5,6]. Traditional management in those cases includes either surgical treatment or percutaneous biliary drainage (PTBD). Both approaches have high technical and clinical success rate, but not negligible complication rate—15% for surgery [7] and up to 33% for PTBD [8]. Additionally surgical treatment is associated with longer hospital stay, while PTBD worsens life quality necessitating everyday care for external drains. In the past decade, with the advent of therapeutic endoscopic ultrasonography (EUS), an alternative to the well accepted techniques emerges called EUS-guided biliary drainage (EUS-BD). In general, EUS-BD consists of ensuring either transpapillary drainage using endosonography for biliary access (EUS-rendezvous (EUS-Rv) or EUS–antegrade stenting (EUS-As)) or transluminal one by creating an artificial fistulous tract in the form of either gastro-biliary (EUS-hepaticogastrostomy (EUS-HGS)) or bilio-enteric fistula (EUS-CDS). Procedure is performed under combined endocopic, ultrasonographic, and fluoroscopic guidance. It obviates the need for external drainage catheters. Furthermore, transluminal approach supposedly results in longer stent patency compared to ERCP. Accumulating data show that EUS-guided drainage is a valid option for biliary decompression with favorable technical and clinical outcomes (over 90% in expert hands) [9]. Herein we present our experience with five cases of obstructive jaundice associated with unresectable pancreatic malignancy treated by EUS-guided drainage, particularly EUS-CDS.

## 2. Patient Selection

Five consecutive patients with obstructive jaundice due to unresectable pancreatic malignancy (either locally advanced or metastatic pancreatic cancer) and treated by EUS-CDS between May 2021 and July 2022 were retrospectively included in the current series. Three of the patients are women and two men. Age interval is between 60 and 74 years. Histological confirmation of PDA, by means of EUS fine needle biopsy (EUS-FNB) or percutaneous Tru-cut biopsy and staging by means of contrast enhanced computer tomography were performed in all patients prior the procedure. All subjects were considered ineligible for conventional endoscopic drainage through ERCP due to either malignant duodenal obstruction (MDO) (type I in 3 patients, type II in 1 patient) or failed cannulation (1 patient). The types of MDO are presented in Table 1. All ERCP interventions were performed by an endoscopist with annual caseload of over 200 procedures. Follow-up was performed until the patient was alive (1–13 months). 

## 3. Ethics

An oral and written informed consent was obtained prior the procedure, with the patients and their relatives being thoroughly informed on the possible clinical outcomes, adverse events, and complications, as well as on the valid alternatives.

## 4. Technique Description

In all cases, thorough blood tests were performed prior the procedure, including complete blood count, CRP, bilirubin, alkaline phosphatase, gamma-glutamyl transferase, amylase, lipase, serum protein, albumin, electrolytes. Abdominal utrasonography was performed before the procedure and the findings were thoroughly recorded. All procedures were performed by an endoscopist performing 300–400 diagnostic and therapeutic EUS per year.

The EUS-CDS was executed under general anesthesia using a combination of fentanyl, midazolam, sevoflurane, suxamethonium (Lysthenon), atracurium besylate (Tracrium), and propofol. All patients received prophylactic antibiotics (ceftriaxone 2.0 g i.v. prior the procedure and at least 3 days after). 

Patient was placed in supine position. A curvylinear echoendoscope (Olympus GF-UCT160-OL5, Olympus, Hamburg Germany) was introduced then and placed at the first part of the duodenum. Insufflation with CO_2_ was utilized instead of ambient air (Olympus UCR). Once in the bulbar part of the duodenum, the common bile duct was identified endosonographically. Color Doppler was used to delineate the surrounding vascular structures and to exclude the presence of major interposing vessel. A crucial step was to ensure that the esndoscope was in such a position so that the needle would be oriented toward the liver hilum. This was achieved by further adjusting the position of the endoscope using fluoroscoping guidance (Philips BV Pulsera C-arm, Philips, Best, The Netherlands). Generally, if on fluoroscopy the endoscope was in a long position and with the tip pointing upwards and facing the insertion tube it was considered the optimal position to access the common bile duct. 19 Ga FNA needle (Expect^TM^ needle; Boston Scientific; Marlborough, MA, USA) was used to puncture the common bile duct. To verify the correct position, bile was initially aspirated, followed by contrast injection (Iopamidol 370 mg/mL) to obtain cholangiogram. Slight irrigation with saline was then performed, followed by insertion of 0.025 inch guidewire (JagWire Revolution^TM^; Boston Scientific; Marlborough, MA, USA), which was then placed in as deep as possible in the intrahepatic bile ducts. Slight withdrawal of the needle was used to avoid “sheering” in case of advancement of the guidewire in the correct direction was cumbersome. Once stable position of the guidewire was achieved, dilation of the fistulous tract was performed using a cystotome (10fr in 1 case, 6fr in 4 cases; Endo-flex GmbH, Voerde, Germany) paired with electrosurgical unit ERBE Vio 300D (Erbe Elektromedicin Gmbh, Tübingen, Germany) set at Endocut I (effect 2, cut duration 3, cut interval 3). Eventuallya fully covered self-expandable metal stent (FC-SEMS) (WallFlex^TM^; Boston Scientific; Marlborough, MA, USA) was inserted—60/10 mm in three cases and 80/10 mm in one case. Every effort was made to place the stent just below the confluence to avoid inadvertent segmental obstruction of the intrahepatic bile ducts. SEMS was intended to extend about 3 cm in the duodenum to reduce the risk of stent migration. After successful positioning of the stent, endoscopic and fluoroscopic evaluation was performed to verify the presence of bile flow and evacuation of contrast media from the bile ducts and to exclude hemorrhage. The described steps of EUS-CDS are depicted in Figure 1. In three patients with concomitant duodenal obstruction, duodenal stenting (WallStent^TM^ 22/120 mm, Boston Scientific; Marlborough, MA, USA) as cointervention was performed during the same procedure. Eventually, conventional US was performed to compare the findings prior and post-procedure. Follow-up US was performed routinely on post-procedure day 1, and lab test results were acquired at day 1 and day 3. During the follow-up, US was performed mandatory at week 2 and week 6 and after that as per necessity. 

The procedure, according to the described protocol, was performed without adverse events and complications and with technical and clinical success in four patients. Technical success was defined as successful creation of choledochoduodenal fistula and placement of FC-SEMS, followed by flow of bile in the duodenum. Clinical success was defined by resolution of jaundice and pruritus and improvement of lab abnormalities (50–75% decrease of bilirubin levels on week 2).

In one patient (who was the first patient in our series) intervention failed. Initially the direction of the needle turned out to be suboptimal since it was pointed perpendicular to the bile duct and more toward the papilla and not the hilum. This led to inability to place the guidewire deep into the intrahepatic bile ducts. An attempt was made then to dilate the fistula with a 10 fr cystotome and reposition the guidewire, but the large bore cystotome turned out to be too rigid to manipulate. Eventually dislocation of the guidewire occurred during manipulation and further access to the bile ducts was impossible as they were almost entirely decompressed. PTBD was not technically feasible, so patient was referred for urgent surgery. Surgical procedure was uneventful with cholecystojejunostomy performed and a T-tube placed at the puncture site, as well as gastroenterostomy for the concomitant duodenal obstruction. Patient had a smooth postoperative period and was discharged on postoperative day 5, and the T-tube was removed on day 7. Throughout follow-up there were no signs of biliary or gastric outlet obstruction. Patient died 4 months later from causes unrelated to the endoscopic or surgical procedure. Patients’ characteristics are presented in Table 2.

During follow-up, there were no cases of stent occlusion or cholangitis. In one patient, obstruction of the duodenal stent due to tumor ingrowth was established 45 days after the procedure. He was further evaluated and was found to have distal duodenal involvement at the region of ligamentum Treitz, so he was referred for surgical bypass.

Current series has some strengths and limitations. Obvious limitations are the small case sample and retrospective selection. On the other hand, it reviews consecutive patients which reduces selection bias. All procedures were performed by single endoscopist, which might suggest the importance of the learning curve and improvement of technique for the clinical outcome.

## 5. Discussion

Since its introduction in the 1980s [10], EUS has rapidly evolved from exclusively diagnostic procedure to therapeutic intervention with enormous capabilities. In the field of bilio-pancreatic diseases, the application of interventional EUS was initially demonstrated by Wiersema et al., who performed the first EUS-guided cholangiography in 1996 [11]. As a pioneer in the EUS-guided biliary drainage (EUS-BD), procedures should be regarded Giovannini who performed the first EUS-CDS in 2001, followed by EUS-guided hepaticogastrostomy (EUS-HGS) in 2003 [12,13]. Since then, EUS-BD has gradually become a well-accepted alternative to standard approaches in obstructive jaundice with published guidelines on its indications, contraindications, and adverse events by ESGE in 2022 [14].

There are three EUS approaches that could be employed to achieve biliary drainage. The first is EUS-Rv in which FNA needle is used to puncture intra- or extrahepatic bile duct, insert a guidewire, and pass it transpapillary in the duodenum. The echoendoscope is replaced by duodenoscope and the procedure is completed like a regular ERCP. An absolute necessity to complete such procedure is the papilla to be endoscopically accessible. An alternative is the EUS-guided transluminal approach, including EUS-CDS and EUS-HGS. It constitutes creation of fistulous tract between the intra- or extrahepatic bile ducts and the stomach or duodenum respectively and placement of stent. This modality has the advantage of not requiring an access to the papilla whatsoever. The third option is EUS-guided antegrade stenting in which a guidewire is passed transpapillary followed by insertion of stent in antegrade fashion through a papilla or anastomosis. While advantageous with respect to preserving normal anatomy, this variation is cumbersome and prone to failures. Recent research show comparable results in terms of technical and clinical success rate for all procedures as follows: 94% and 88% for EUS-CDS, 96% and 87% for EUS-HGS, 86–95% and 77–95% for EUS-As, and 84% technical success rate for EUD-Rv [15,16,17,18]. In terms of AE, performance is also comparable with slight advantage of EUS-CDS over EUS-HGS and EUS-Rv [19,20]. The better safety profile determines EUS-CDS to be preferable to EUS-HGS for malignant distal obstruction particularly in patients without proximal duodenal involvement or altered anatomy. The presence of large volume ascites is a major issue for any EUS-BD, but especially for EUS-HGS. Summary of the indications for performing EUS-BD is presented in Figure 2. 

The main indication for EUS-CDS is malignant distal biliary obstruction (predominantly pancreatic cancer) as is the case in our study. The technique utilized is quite similar between studies differences lying mainly in the type of guidewire (0.025 inch or 0.035 inch), the accessories used for dilation of the fistulous tract, and the type of stent inserted. The type of anesthesia and gas used for insufflation are also still debatable, though general understanding being that general anesthesia and CO_2_ insufflation are compulsory [21]. Usage of 19 Ga is generally accepted, while both 0.035 inch and 0.025 inch guidewires could be used. In our series, we used 0.025 inch JagWire Revolution (Boston Scientific; Marlborough, MA, USA). To our knowledge, this is the first time, the usage of this particular guidewire is mentioned in the literature. Its enhanced stiffness compared to the standard JagWire^TM^ ensures better ratio between stiffness and mobility, making maneuvering through the needle without “sheering” easier [21]. Furthermore, upon stent introduction, increased rigidity of the guidewire facilitates insertion. In our opinion, advancing the guidewire deep into the intrahepatic bile ducts is the decisive factor for a successful procedure, so further studies evaluating the optimal guidewire for the intervention are justified. In our series, we used combined ultrasonographic and fluoroscopic guidance to align the echoendoscope toward the liver hilum which was considered a crucial prerequisite for subsequent guidewire advancement. To date in the literature a detailed description of this approach is hardly provided, and we consider it to be of high practical value for most interventional endoscopist. The type of accessory used for dilation is possibly the most debatable part of the EUS-BD technique. Failure to dilate the fistula is the chief cause for technical failure, and overdilation could cause bile leakage as was in our case. Mechanical dilation using bougie or balloon dilation is recommended, while usage of non-coaxial electric cautery (needle knife) is generally disregarded, because of not negligible bleeding risk [21,22]. In our series, we did not use mechanical dilators since we feared guidewire dislocation. Instead, we used coaxial cautery catheters (cystotome by Endo-flex GmbH) which are considered an option in current guidelines [23]. We experienced no cases of bleeding and access was obtained fairly easily which might suggest that this technique is safe and feasible and urges further investigation. We would consider 6 fr cystotome safer since maneuverability is better and the duodenal wall defect is much smaller compared to the 10 fr cystotome we used in our first case. In the early days of EUS-BD, both plastic stents and SEMS were used. Nowadays usage of plastic stents is largely abandoned due to high AE rate [22,23,24]. FC-SEMS are predominantly the choice in EUS-CDS. A 4 cm stent might be used, but due to the risk of stent dislocation, 6 cm or even 8 cm stents are more commonly used. In our series we used FC-SEMS 6 cm (3 patients) and 8 cm (1 patient) with the length being determined mainly by the patient’s anatomy, particularly the distance from the puncture site to the hepatic confluence. More recently the usage of lumen apposing metal stents (LAMS) (HOT-AXIOS, Boston Scientific, Marlborough, MA, USA) for obtaining biliary drainage was investigated. LAMS have the advantage of being able to convert a complex multistep procedure to a single-step one. Additionally, their design ensures the creation of a stable and sealed fistula between the duodenum and CBD, thus reducing the risk of bile leakage. The clinical and technical success rate of EUS-CDS using LAMS was excellent in a study by Anderloni et al. with 11.6% rate of major AE, which is generally lower compared to EUS-BD AE in general (17–23%) [15,25]. Certain concerns regarding the bleeding risk still exist [26]. They derive mainly from the increased rate of bleeding complications when using LAMS for drainage of pancreatic fluid collections. Whether this finding is attributable to EUS-BD is still debatable. Overall small diameters (6–8 mm), LAMS are a good option for EUS-CDS, though cost effectiveness is still an issue.

In our series, adverse events were established in one patient during our first experience with the procedure. We might hypothesize that learning curve would inevitably influence technical success and AE rate, which is supported by a study of Poincloux et al., who established a mortality rate of 10% in their first 50 procedure, after which mortality dropped to 2% in the next 50 [27].

According to recent research, EUS-BD might replace PTBD in cases of failed ERCP and eventually even replace ERCP as first choice for patients with malignant biliary obstruction. This is associated with the lower incidence of some ERCP-specific complications such as pancreatitis, stent occlusion, and cholangitis [17,28]. The experience in our center, performing about 50 PTBD procedures annually, also suggests that EUS-BD may have better safety profile and clinical outcome compared to PTBD. Regarding EUS-BD as a first-choice modality, current research advocate it only in high volume centers, a statement we strongly support [14].

## 6. Conclusions

Treatment of obstructive jaundice in pancreatic cancer is one of the main steps in the complex management of the disease. ERCP is an accepted first line treatment, which though inevitably fails in some patients. PTBD and surgery are valid alternatives, but they are associated with not negligible morbidity. EUS-CDS is an emerging technique in patients with malignant distal biliary stenosis. In the current series, certain technical aspects of the procedure such as accurate alignment of the endoscope toward the liver hilum and optimal choice of guidewire and dilation device were outlined. Our initial experience with the procedure suggests that it is with high technical and clinical success rate and acceptable safety profile even when performed in medium volume endoscopic center. Further studies comparing EUS-CDS and PTBD might be of use for the everyday clinical practice. 

## Figures and Tables

**Figure 1 medicina-59-00351-f001:**
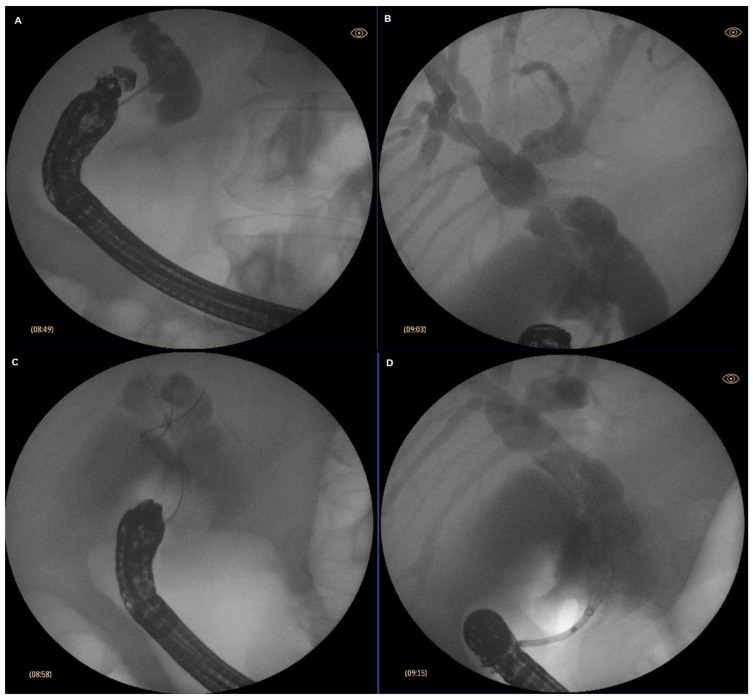
Stages of endocscopic ultrasound-guided choledochoduodenostomy (EUS-CDS); (**A**) puncture of the common bile duct (CBD); (**B**) advancement of the guidewire; (**C**) dilation of the fistulous tract; (**D**) stent insertion.

**Figure 2 medicina-59-00351-f002:**
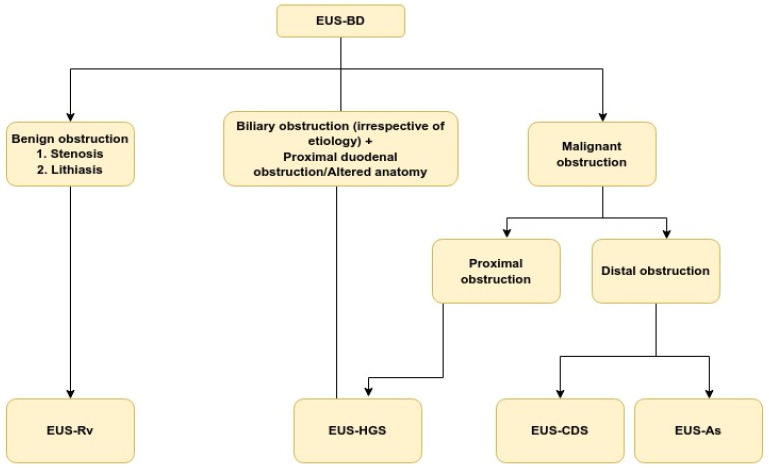
Summary of current indications for EUS-BD. EUS-BD: Endoscopic ultrasound-guided biliary drainage; EUS-RV: EUS-guided rendezvous technique; EUS-As: EUS-guided antegrade stenting; EUS-CDS: EUS-guided choledochoduodenostomy; EUS-HGS: EUS-guided hepaticogastrostomy.

**Table 1 medicina-59-00351-t001:** Types of malignant duodenal obstruction.

Types of MDO	Level of the Obstruction
Type I	Proximal to duodenal papilla
Type II	At the level of duodenal papilla
Type III	Distal to duodenal papilla

MDO—malignant duodenal obstruction.

**Table 2 medicina-59-00351-t002:** Patients’ characteristics.

Patient	Sex	Age	Cause for ERCP Failure	Technical/Clinical Success	Complications	Blood Tests Prior—1 Day after Procedure				
						WBC mm^3^	CRP U/L	Bil. µmol/L	AP U/L	GGT U/L
1	F	74	Duodenal obstruction	No/No	Perforation/Bile leakage	8.2–11.4	34.1–54.4	186.1–97.4	607–701	478–390
2	M	70	Duodenal obstruction	Yes/Yes	No	7.2–6.5	11.4–41.1	99.4–69.7	493–399	927–679
3	F	61	Duodenal obstruction	Yes/Yes	No	10.1–8.8	5.1–37.3	164.5–121.0	272–228	586–274
4	F	60	Papilla invasion	Yes/Yes	No	8.3–6.0	15.3–15.4	252.0–73.6	585–448	912–685
5	M	73	Duodenal obstruction	Yes/Yes	No	13.3–9.0	8.7–12.2	343.2–211.0	2003–1679	891–695

ERCP—endoscopic retrograde cholangiopancreatograhy; WBC—white blood cells; CRP—C-reactive protein; Bil.—bilirubin; AP—alkaline phosphatase; GGT—gamma-glutamyl transpeptidase.

## Data Availability

Not applicable.

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
