# Peer review of "Clinical Outcomes of EUS-Guided Choledochoduodenostomy for Biliary Drainage in Unresectable Pancreatic Cancer: A Case Series"

_medicina, 2023, doi:10.3390/medicina59020351_

Round 1

Reviewer 1 Report

Dear authors,

Thank you very much for giving me the opportunity to review the paper entitled ‘Clinical outcomes of EUS-guided choledochoduodenostomy (EUS-CDS) for biliary drainage in unresectable pancreatic cancer: A case series. I have several comments to the authors.

As authors mentioned in the manuscript (Discussion, first paragraph), EUS-guided biliary drainage (EUS-BD) including EUS-CDS was initially performed over 20 years ago, and now those procedures are well-established among pancreatobilialy endoscopists in high-volume centers. I could not find any novel things from these 5 cases in the paper. In my opinion, I recommend authors to clarify the novel things.

Thank you.

Reviewer 2 Report

In this article, the authors described their experiences of five cases treated with EUS-CDS for their biliary obstruction due to unresectable pancreatic cancer. They succeeded in four cases within five cases of biliary obstruction by pancreatic cancer. 

The technique of EUS-CDS was interesting and looks useful for obstructive jaundice. 

Instead of this technique, EUS-HGS is also widely used for biliary obstruction due to periampullary malignancy. The authors should discuss more regarding the comparison between EUS-CDS and -HGS. They just showed the success rate and AE rate of these two techniques. They should discuss the advantages and disadvantages of both methods and show the adequate cases in which EUS-CDS should be selected for biliary drainage.

Reviewer 3 Report

Despite the low number of patients in this study, the EUS-BD is an interesting topic and case series about this field are appreciated. The authors should be more concise in the introduction section. The procedures are well described. Since the introduction of dedicated accessories and prostheses for EUS-BD made these procedures more successful with regard to technical success, clinical outcomes, and reduction of adverse events (AEs),  LAMS should be mentioned in the discussion.

Round 2

Reviewer 1 Report

Thank you very much for the revise. I understand the merit of the paper.